# Maximally efficient prediction in the early fly visual system may support evasive flight maneuvers

Siwei Wang[1,2], Idan Segev[3,4], Alexander Borst[5], Stephanie Palmer[1,2]*

**1** Department of Organismal Biology and Anatomy, University of Chicago, Chicago, Illinois, United States of America, **2** Department of Physics, University of Chicago, Chicago, Illinois, United States of America, **3** Department of Neurobiology, Hebrew University of Jerusalem, Jerusalem, Israel, **4** Edmond J. Safra Center for Brain Sciences, Hebrew University of Jerusalem, Jerusalem, Israel, **5** Max Planck Institute of Neurobiology, Martinsried, Germany

* sepalmer@uchicago.edu

**Data Availability Statement:** The paper is a theoretical work and does not contain experimental data. All the parameters and open source software packages required to reproduce our simulation and

## Abstract

The visual system must make predictions to compensate for inherent delays in its processing. Yet little is known, mechanistically, about how prediction aids natural behaviors. Here, we show that despite a 20-30ms intrinsic processing delay, the vertical motion sensitive (VS) network of the blowfly achieves maximally efficient prediction. This prediction enables the fly to fine-tune its complex, yet brief, evasive flight maneuvers according to its initial ego-rotation at the time of detection of the visual threat. Combining a rich database of behavioral recordings with detailed compartmental modeling of the VS network, we further show that the VS network has axonal gap junctions that are critical for optimal prediction. During evasive maneuvers, a VS subpopulation that directly innervates the neck motor center can convey predictive information about the fly's future ego-rotation, potentially crucial for ongoing flight control. These results suggest a novel sensory-motor pathway that links sensory prediction to behavior.

## Author summary

Survival-critical behaviors shape neural circuits to translate sensory information into strikingly fast predictions, e.g. in escaping from a predator faster than the system's processing delay. We show that the fly visual system implements fast and accurate prediction of its visual experience. This provides crucial information for directing fast evasive maneuvers that unfold over just 40ms. Our work shows how this fast prediction is implemented, mechanistically, and suggests the existence of a novel sensory-motor pathway from the fly visual system to a wing steering motor neuron. Echoing and amplifying previous work in the retina, our work hypothesizes that the efficient encoding of predictive information is a universal design principle supporting fast, natural behaviors.

results are specified in this Materials and Methods section. We prepared a github repository https://github.com/siwei-wang/VS_pred including all intermediate data and code we used to generate figures in the manuscript.

**Funding:** This work was supported in part by the Gatsby Charitable Foundation (https://www.gatsby.org.uk/neuroscience) (SW, IS) and by the Max Planck Hebrew University Center for Sensory Processing of the Brain in Action (https://www.mpg.de/7021540/hebrew_uni_center_Jerusalem) (SW, IS, AB). The work was also support by the National Science Foundation, both via CAREER award 1652617 (SEP, SW), and through the Center for the Physics of Biological Function (PHY-1734030) (SEP), and by the National Institutes of Health BRAIN-R01 EB026943 (SEP, SW). The funders played no role in the study design, data collection and analysis, decision to publish, or preparation of the manuscript.

**Competing interests:** The authors have declared that no competing interests exist.

# Introduction

The goal of sensory processing is to guide behavior. Whether we are trying to catch prey, escape a predator or find food within a complex environment, the data our senses collect are only useful to the extent that these data can improve the success of our future actions. However, most previous work [1, 2] that established that early sensory processing might be optimal only applied the notion of optimality to the physical limits of sensing, or the statistics of the external input alone. To move beyond this picture, one must consider what sensory information is used for. For this problem, the fly is an ideal testbed: both precise measurements of its behavior and detailed mechanistic understanding of its underlying sensory processing are available, so one can investigate behavioral constraints that arise from key survival strategies and how these constraints might sculpt its early sensory circuits.

Here we focus on how targeted sensory processing in the fly may support one key survival-critical behavior: in-flight evasive maneuvers. To maximize the animal's survival, escapes generally take place at strikingly fast timescales [3]. Such short timescales are usually comparable to the timescale of the inherent sensory processing delay. For example, *Drosophila* can finish its evasive maneuver within a mere 40ms after the detection of a purely visual threat. This is at the same timescale of its visual processing delay, which is about 30ms [4, 5]. Because of this, previous work has suggested that escape behaviors are largely planned, i.e. that they unfold as a "motor tape". However, while visual information may not be updated during the course of the escape maneuver, it is still possible that the initial visual information can *inform* how the escape unfolds. These in-flight evasive maneuvers are banked turns, i.e. initiated with a rotation immediately followed by actively damping of that turn to start counter-rotation in the opposite direction. In addition, the maneuvers themselves require active and on-going control: i) The initial rotation is not stereotyped (i.e. purely 90˚ or on predetermined body axes as are saccadic flight turns [6]). It must be tuned to both the position of the looming threat and the animal's heading at the moment of detection. ii) The subsequent counter-rotation determines how successful the animal is at turning away from the threat by changing its prior flying trajectory. Thus, it also needs to be fine-tuned in an ongoing manner instead of being a simple correction of the initial rotation. iii) Both the initial rotation and the follow-up counter rotation (and their combined trajectories) must be variable even for visual threats appearing on the same position to avoid the predator's trap (e.g. an electric snake can mitigate its prey's predictable, reflexive C-start [7] and thus foil the escape). Previous experiments identified that the mechanosensory circuit, i.e. the halteres, can perform such motor control for voluntary saccadic turns. Thus, it was hypothesized that the halteres are also responsible for the motor control during the escape.

However, the banked turns of evasive maneuvers are five times faster than those observed in the voluntary saccadic turns. The halteres alone do not have sufficient dynamic range to sense these fast rotations. Halteres are known to be gyroscopic sensors [8]; they help a fly keep its aerodynamic balance [9]. Like the vestibular system, they respond to short timescale rotational perturbations (e.g. air anisotropies) and experience the Coriolis forces during a fly's body rotation. Previous work showed that the halteres can sense ego-rotation with angular velocities up to 2500˚/*s* [10, 11]. Because voluntary saccadic turns, i.e. fixed stereotyped rotations, have angular velocities around 1000˚/*s* only, most previous studies are able to use these voluntary saccadic turns to show how the halteres control behavior. Nevertheless, this dynamic range is dwarfed by the angular velocities achieved during evasive maneuvers, which can be as high as 5300˚/*s*. Alternatively, descending visual inputs can shift the dynamic range of the halteres, which allows them to engage in active control when encouraging high angular velocity during evasive maneuvers. This is the so-called control loop hypothesis [12] and

experimentally validated recently [13]. Such vision-mediated control has also been observed in other behaviors [14–16].

The descending visual inputs useful for haltere control are the global motion responses generated by the large tangential lobula plate cells (LPTC) in the fly visual system. The fly visual system is organized in four largely feedforward layers: retina, lamina, lobula, and lobula plate. Located at the 4th neuropil, these LPTCs have a roughly 30 ms processing lag across many dipterans [4, 5, 17]. Because of this processing lag, it is unlikely that evasive maneuvers can access visual information through feedback during the escape. Instead, we propose that the visual information of ego-motion at the time of threat detection may play a new feedforward role. Said another way, by using its own past visual input before the evasive maneuver to predict the its future visual experience during that maneuver, a fly bypasses this processing lag and still uses visual inputs for active control. This so-called "bottom-up" prediction exploits the temporal correlations between past and future visual stimuli (due to the stereotypy in the escape behavior) during evasive maneuvers. Such bottom-up prediction exists in the vertebrate retina [18, 19], and ensures fluid interaction with the external environment. It was also shown to be important in the formation of long-term memory [20]. In our case, the escape trajectory depends on the threat angle relative to the fly's heading [21]. Where and how the escape maneuver begins constrains how it will unfold, giving the visual system ample predictive power with which to feed forward into active flight control. We show how this bottom-up prediction provides information about the future sensory input, subverting delays in the visual input stream. Because blowflies are known to execute higher velocity saccadic turns than *Drosophila*, this bottom-up prediction will be even more important for blowflies' shorter and faster evasive maneuvers.

The neural architecture of insect brains is highly conserved. It is hypothesized that the neural circuits and escape behaviors co-evolved, as early as flight itself [22]. Many modern arthropod species inherited these core sensory-behavioral modules. Both the blowfly and *Drosophila* use banked turns to change their heading direction [6, 23, 24]. Although blowflies in general execute higher velocity saccadic turns and have higher acuity in their compound eyes than *Drosophila*, both animals share similar wingbeat frequencies [25, 26]. This suggests similar time courses in their escape trajectories. Because of the high angular velocity, banked turns during evasive maneuvers are mostly composed of pitch/roll combinations. These rotations generate rotational flow-fields processed mainly by the vertical system (VS) network [27–29]. The VS cells in both the blowfly and *Drosophila* have analogous electrotonic structures [30] despite their size difference (a blowfly is roughly four times bigger than a *Drosophila*). Similarly, their mechanosensory and motor systems are scaled according to this size difference [13, 25, 31–36]. Because the only precise measurements of the fly's evasive maneuver are available in *Drosophila* [21] and the only experimentally validated neural circuit that processes the visual input during these maneuvers is available from the blowfly [37–39], but the two species are so similar overall [40], we investigate how the blowfly's motion sensing circuit extracts behaviorally relevant information based on behavioral measurements from *Drosophila*.

Although both the local motion detection at the retina [41, 42] or the global motion processing at the VS network have long low-pass time constants of 150-200 ms to achieve its optimum temporal frequency, these considerations are specific in the steady state. Previous experiments [43] and modeling [44] showed that single VS cells elicit transient detector responses for short abrupt maneuvers at the timescale of milliseconds. This makes the VS network suitable for evasive maneuvers. In addition, recent theoretical work [45, 46] showed that the population coding scheme in the VS network efficiently encodes constant speed rotational motion using transient response only, i.e. the output of the initial 10 ms after encountering a motion stimulus. This effect was dependent on the positioning of gap junctions between the

axons of neighboring VS cells, and has been experimentally validated in [37, 47]. This work suggests that the VS network can encode motion information on a timescale compatible with evasive maneuvers. However, without determining if the output of the encoding scheme is behaviorally relevant, this cannot determine if such processing is useful for downstream motor control. In naturalistic behaviors, ego-motion is highly dynamic. Encoding a past ego-motion is only useful if it is informative for the animal's future experience. Merely maximizing information about the instantaneous motion is not the challenge the animal faces during evasive maneuvers; the VS network must extract the *predictive information* about future motion. In order to identify whether the specific wiring architecture of the VS network supports evasive maneuvers, we ask whether it encodes predictive information about future ego-motion using within its own transient responses.

To explore this hypothesis, ideally one would trace the activity of the VS cells in behaving animals. However, evasive flight maneuvers require untethered flight, which makes population recording from the VS network prohibitive. Furthermore, it is not feasible to block the VS network in flying animals, because they are essential for optomotor responses [48, 49]. Therefore, we use numerical simulations of an in-silico, biophysically realistic compartmental reconstruction of the VS network to investigate how the VS network might encode this kind of fast, predictive information. This compartmental reconstruction of the VS network is highly experimentally constrained [37, 38]. All single-cell [50] and neuronal circuitry parameters [37, 51, 52] are set such that this compartmental reconstruction behaves as does the real VS network when presented with the same current injection [37, 38, 51, 52]. Our computational approach (using a model when large scale recordings from a complete circuit during natural behavior is not possible) is similar to how previous work on electric fish [53] used a synthetic population model of the electrosensory lobe to show how accurate cancellation of self-generated electric signals is achieved.

We first show that the VS network uses axonal gap junctions to output substantial predictive information for evasive maneuvers. Next, we show that this predictive information is near-optimal throughout the duration of evasive maneuvers—it can be used to prospect forward in time with equally fidelity through the escape. We further show that the output circuitry of the VS network (the VS 5-6-7 triplet) to the neck motor center retains all available information about the future stimulus, i.e. compressing the readout does not sacrifice how much a downstream pathway knows about the ongoing evasive maneuver. Finally, we identify that the encoding of predictive information is particularly suitable to fine-tuning future motion directions. These results suggest possible novel sensory-motor pathways: either a direct connection between the lobula plate descending neurons to the wing steering muscles [54–56], or an indirect connection from the visual system through the halteres to the wing steering muscle, as proposed in the control loop hypothesis [12, 13].

## Results

### Visual prediction provides substantial information about motion without delay

We show in Fig 1 that visual prediction contains substantial information about future motion that may be important for controlling evasive flight maneuvers. We first use a schematic trace to illustrate the inputs and delays in the fly visual system (Fig 1A). Previous works showed that the haltere outputs reach the wing steering muscles after a 15-20 ms delay [21, 57], towards the second half of the maneuver, right before the active counter-banked turn starts. Visual feedback would arrive too late, coming online only after 30 ms, long after the initial rotation is replaced by the counter rotation through active control [57].

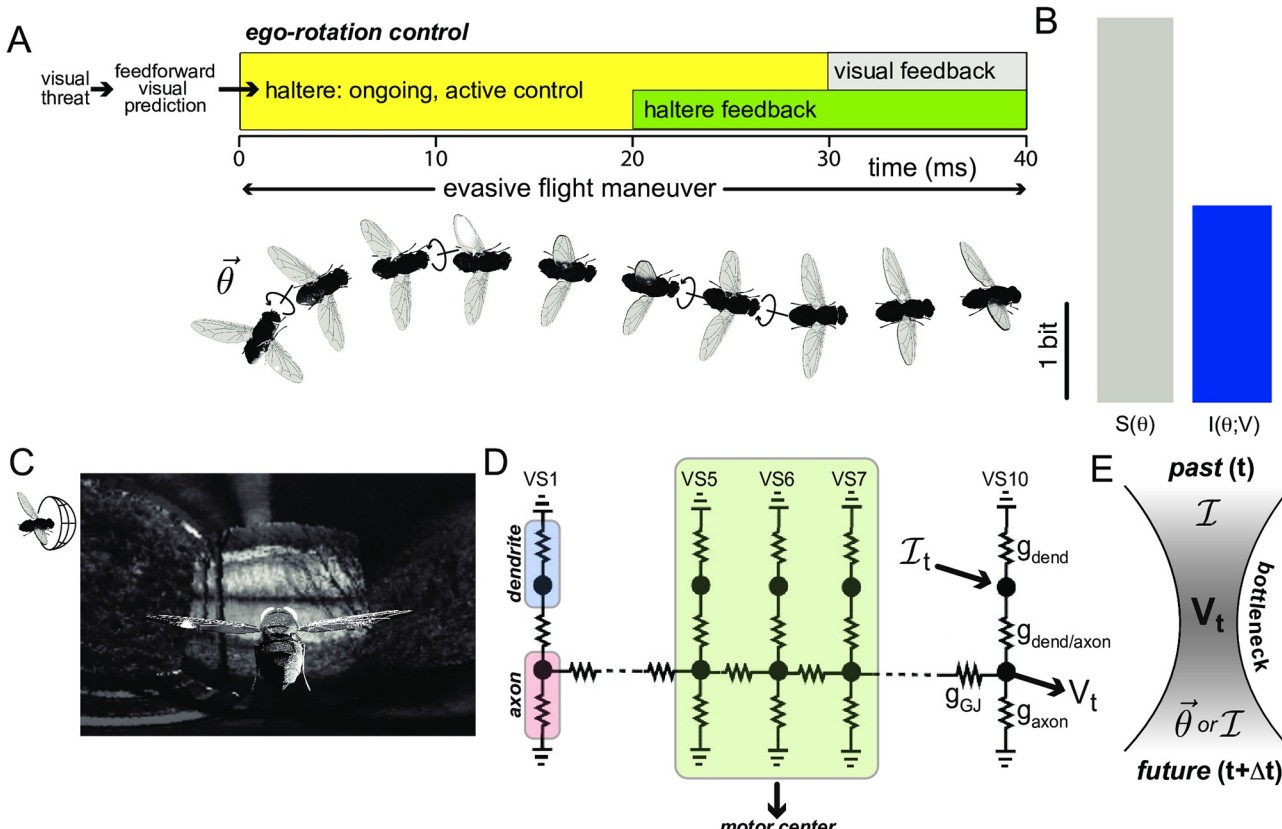

**Fig 1. Predictive information is the dominant information source about visual inputs during evasive flight maneuvers.** (A) Upon emergence of a threat (shown as the red star), dashed arrow represents the visual-motor delay of 60 ms from the onset of threat to the start of evasive maneuvers. After this sensory-motor delay, the position of the threat is known. The fly performs an evasive maneuver by changing its heading through a banked turn (arrows show a rotation at direction $\vec{\theta}$ and its respective counterrotation). During evasive maneuvers, visual predictions can provide motion information throughout the entire duration, i.e. without delay (shown as the yellow zone), whereas the haltere feedback is only available after 20 ms (shown as the green zone) and the visual feedback is only available after 30 ms (shown as the shaded zone). The arrow leading to the haltere system illustrates how visual information might regulate haltere activity (as recently shown in [13]): because of the 30 ms sensory processing lag, haltere activity must be regulated by visual prediction. (3D fly rendering courtesy of D. Allan Drummond.) (B) This histogram compares how much information the visual prediction (shown in blue) can encode about ego-rotation ($I(\theta, V)$) during evasive maneuvers with their respective entropy (shown as $S(\theta)$ in gray). We use the ego-rotation distribution at $\Delta t = 10ms$ into the evasive maneuver to compute this entropy. Its distribution is shown in S2(A) Fig. Note that the VS output encodes almost half of the entropy of a future ego-rotation. (C) The Mercator map of a randomly generated natural scene background. To generate this map, we first randomly generate a natural scene environment. We then generate a movie mimicking an evasive flight in the natural environment by rotating this natural scene environment according to the respective measured rotations. We project this movie to a unit sphere that represents the fly's retina, see details in S1 Fig. There are 5,000 local motion detectors (LMD) on this unit sphere as on the fly's retina. The responses of these LMDs are then integrated as the input current $\mathcal{I}$ to the VS network (shown as an arrow to D). (D) A biophysically detailed model of the VS network, based on known neural circuitry [50, 51]. Note that because the soma is decoupled in VS cells (only connecting to the rest of the cell via a soma fiber), we leave out the soma in this VS model. We highlight the outputs to the neck motor center here, the axonal voltages of the VS 5-6-7 triplet. This is the only known readout that directly connects to motor pathways. (E) A cartoon showing how the information bottleneck problem is setup for prediction in this system: using the general correlation between a past input (either the ego-rotation $\vec{\theta}$ or the corresponding dendritic input $\mathcal{I}$, this information bottleneck finds a compact representation $V$ (i.e. the bottleneck defines how much information about the input is 'squeezed out' when $V$ is generated) of the past (the past input to the VS network: $\mathcal{I}_{past}$) that retains predictive components about the future ($\theta_{future}$ or $\mathcal{I}_{future}$).

To quantify how much visual prediction encodes about ego-rotation ($\theta$) in the fly's future escape trajectories (Fig 1B), we define this ego-rotation-relevant predictive information in the output voltage from the fly VS network as $I_{future}(\theta, \Delta t)$ (Eq 1, abbreviated as $I_{future}^{\theta}$),

$$
\begin{aligned}
I_{future}(\theta, \Delta t)(\Delta t) \quad &= I(V_{past}; \theta_{future}) \\
&= I(V_t; \theta_{t+\Delta t}),
\end{aligned} \tag{1}
$$

where $V_t$ is the output axonal voltage of the VS network at time $t$. $\Delta t$ is the time interval between the past voltage and future ego-rotation. Here, we use intervals of $\Delta t = 10ms$, $20ms$, $30ms$, $40ms$ to obtain the output of the VS network. This is because the maximum firing rate of the descending neuron connecting to the neck motor center is 100 Hz [38], which corresponds to an integration step of at least 10 ms (see Materials and methods). Throughout this paper, we represent future ego-rotations $\theta_{t+\Delta t}$ by their vector components $(\cos(\theta_{t+\Delta t}), \sin(\theta_{t+\Delta t}))$. The cosine component corresponds to roll direction/magnitude and the sine component corresponds to pitch direction/magnitude. This vector is within the fly's coronal plane, to which the VS neurons are selectively sensitive. We then estimate $p(\theta_{t+\Delta t})$, i.e. the stimuli distribution and $p(\theta_{t+\Delta t}|V_t)$, i.e. the probability of future ego-rotation conditioned on the past output axonal voltage to obtain $I_{\text{future}}(\theta, \Delta t)$ (see Materials and methods). Fig 1B shows that the predictive information $I_{\text{future}}(\theta, \Delta t)$ in the VS output voltages captures nearly 50% of the entropy of the future escape trajectory. This is because where and how the escape maneuver begins constrains how it will unfold, giving the visual system ample predictive power. This suggests that the predictive information encoded by the VS network is an important information source for evasive flight behaviors in the natural environment. This also supports that the haltere is a multisensory integration circuit, other sensory modalities, e.g. antenna [58, 59], ocelli [60–62] or wing reflexes [63, 64] may provide additional motion information.

In the original paper describing the escape trajectories of *Drosophila* [21], the authors found that escape trajectories remain consistent despite different expansion velocities and expansion angles of the visual threat. Therefore, the specific visual features of the threat were not important. These authors also used a large range of sizes for the spatially uniform visual threat. A visual threat as large in diameter as their maximum expansion angle (107˚) is similar to a uniform object found in a natural scene, like a large patch of sky or other large, low-contrast object. Thus, we hypothesize that using a generic natural scene background will not significantly change our results.

To evaluate $I_{\text{future}}(\theta, \Delta t)$, we need to approximate both ego-rotation distributions and the respective output distributions of the VS network. To obtain these ego-rotation distributions, we generate 650,000 samples of visual field motion experience based on behavioral recordings published in [21]. Each visual experience corresponds to one instance of a particular evasive maneuver embedded in a randomly selected set of nature scene images. There are 10,000 samples for each of the 65 evasive flight trajectories with duration of 40 ms (out of the total 92 published trajectories in [21]). Fig 1C shows one exemplar visual experience of a particular evasive maneuver trajectory. Here, we obtain the "cage" of natural images for simulation by randomly selecting six images out of the van Hateren dataset [65] and patch them onto the six faces of a cube. Then we generate a movie mimicking an evasive flight in the natural environment by rotating this natural scene cage according to measured rotations in an evasive flight trajectory S1 Fig (Because previous work [39] showed that the VS network is not responsive to translation motion, we do not use the translation component of evasive maneuvers in this simulation, also see Materials and methods). We next project this movie onto a unit sphere that represents the fly's retina, following the protocol introduced in [39, 45]. There are 5,500 local motion detectors located on this unit sphere, whose outputs are local motion estimates based on pixel intensity difference between neighboring photoreceptors.

To evaluate whether the VS network encodes $I_{\text{future}}(\theta, \Delta t)$ efficiently, we need to define a few additional information theoretic quantities within the VS network architecture (Fig 1D). The dendrites of all VS cells receive current inputs resulting from integrating the outputs from hundreds of upstream local motion detectors [51]. These outputs are then integrated and become the input current $\mathcal{I}$ to the VS network (Fig 1D), which outputs axonal voltages $V$ for downstream processing. Because of the feedfoward structure in the fly visual system (retina,

lamina, lobula, lobula plate), the VS network, located at the 4th neuropil the lobula plate, does not have direct access to the visual input. Therefore, we need to define a proxy for correlations between past and future ego-rotations based on the past and future input currents. We probe how the VS network might use these correlations to encode predictive information about the future ego-rotation. In this encoding scheme, the generalized correlation between the past and the future of the input current $I_{\text{future}}^{\max}$ (Eq 2) itself limits how much predictive information the VS network can encode:

$$
\begin{aligned}
I_{\text{future}}^{\max}(\mathcal{I}, \Delta t) = &\quad I(\mathcal{I}_{\text{past}}; \mathcal{I}_{\text{future}}) \\
= &\quad I(\mathcal{I}_t; \mathcal{I}_{t+\Delta t})
\end{aligned}
\tag{2}
$$

This is the mutual information between the past and future input (the dendritic current) and defines the total interdependence of the current with itself in time.

Similar to $I_{\text{future}}^{\max}$, we define how much information is retained by the axonal voltage of the VS network about its future input as $I_{\text{future}}$ (Eq 3, abbreviated as $I_{\text{future}}^{\mathcal{I}}$).

$$
\begin{aligned}
I_{\text{future}}(\mathcal{I}, \Delta t) = &\quad I(V_{\text{past}}; \mathcal{I}_{\text{future}}) \\
= &\quad I(V_t; \mathcal{I}_{t+\Delta t}).
\end{aligned}
\tag{3}
$$

This is the predictive information between the output axonal voltage and the future input current, which again we are using as a proxy for future ego-rotation.

All of the information encoded by the VS network comes from its sole input current, $\mathcal{I}_{\text{past}}$. To quantify the efficiency of encoding, we need to qualify not only the prediction (i.e. the $I_{\text{future}}$), but also determine how much the axonal voltages consumes to obtain the predictive information, i.e. how much they encode from the input at the same past time. This is mutual information quantity: $I_{\text{past}}$ (Eq 4),

$$
\begin{aligned}
I_{\text{past}} = &\quad I(V_{\text{past}}; \mathcal{I}_{\text{past}}) \\
= &\quad I(V_t; \mathcal{I}_t).
\end{aligned}
\tag{4}
$$

Because brief and fast evasive maneuvers have a quickly varying ego-rotation, we identify prediction as the most important 'relevant' variable in the input to the VS system and set up the corresponding information bottleneck problem (Fig 1E, also see Materials and methods): this method finds the maximal amount of relevant predictive information that the VS network can encode about the future ego-rotation $\theta_{\text{future}}$ and its proxy, i.e. the input $\mathcal{I}_{\text{future}}$, via its axonal voltages at a past time, $V_{\text{past}}$. The solutions to the information bottleneck problem, $I_{\text{future}}^*(\mathcal{I}, \Delta t)$ or $I_{\text{future}}^*(\theta, \Delta t)$, are subject to a constraint on the amount of information $V_{\text{past}}$ has about the input in the past, $I(V_t; \mathcal{I}_t)$. The absolute maximum of the predictive information is set by the generalized correlation between past and future of the ego-rotation trajectory itself, $I(\theta_{\text{past}}, \theta_{\text{future}})$, (and its encoding in the VS network input, $I_{\text{future}}^{\max}$), which we take from real fly maneuvers.

The specific texture of the visual scene significantly impacts the ego-motion-induced input to the VS dendrites. Of course, if the fly rotates against a uniform background, no visual motion information is available. The details of the spatial correlation structure in the scene also affect the input to the VS system and its subsequent ego-rotation estimate. Previous work [37] compared motion encoding using two different visual backgrounds: an artificial background with regular texture (e.g. random dots) and a natural scene background with irregular texture and an inhomogeneous contrast distribution. Those results showed that with the artificial background, rotational optic flow can be reliably read out at the VS dendrites. Axonal gap junctions do not improve this estimate. However, the irregularly distributed patches of high

and low contrast in natural scenes make the motion estimation noisy, even after the dendritic integration. Only once the outputs are filtered through the VS axons with gap junction coupling does robust encoding of the fly's rotational optic flow emerge. By pooling inputs across neighboring VS cells, axonal gap junctions average out the fluctuations in local motion measurements due to irregular textures in natural scenes. It was identified that such contrast normalization denoise the dendritic inputs to the VS network [66], recent work further showed that this contrast normalization acts nearly instantly [67, 68], compatible for the processing during evasive maneuvers. Moreover, [46] showed that even with those axonal gap junctions, the VS network can only encode 50% of the motion information in natural scenes, as compared to artificial, regular textures (shown in S1(C) Fig). Here we are challenging our *in silico* fly with both naturalistic self-motion and complex visual texture. The axonal gap junctions in the VS system may have evolved to solve this particular, behaviorally relevant problem in natural scenes.

## Axonal gap junctions enable prediction during evasive maneuvers

The VS network generates wide-field rotational field responses combining upstream vertical local motion detection [27–29]. There are 20 different VS cells (10 at each compound eye). VS dendrites integrate upward and downward local motion signals from T4 and T5 such that their axons output global motion components of ego-rotation based on the fly's visual experience. Each VS cell has its receptive field centered at a specific rotational axis of the fly's coronal plane (combinations of all pitch and roll rotations). They are numbered VS1-VS10 according to their preferred ego-rotation arranged along the fly's anterior-posterior axis [69]. Recordings from VS cells confirm that they mainly respond to rotational flow fields but not expanding flow fields [49]. If the animal is moving in an open field, translational motion does not generate rotational flow, and the VS network is only sensitive to ego-rotation. VS cells are upstream of both other LPTCs (previous work described ultra-selective LPTCs for detecting the size and speed of a looming target (LPLC2 and LC4 [70, 71]) as well as descending neurons connecting to the neck motor center. Because the rotations during evasive maneuvers are mainly pitch and roll combinations (similar to how a fighter jet gains tactical advantage), the VS network is a good candidate for encoding information specific to and most relevant for this behavior.

Previous work identified the $R_m$, $R_a$, $C_m$, $g_{exc}$, $g_{inh}$ for each VS cell [50] and observed that the network performance of the VS network is dominated by its chain-like wiring architecture [37, 50–52]. Each VS cell only connects with other VS cells with immediately neighboring receptive fields, e.g. VS2 only connects to VS1 and VS3. Meanwhile, the VS1 and VS10 cells show reciprocal inhibition [72]. Previous dual-recording experiments [51] have shown that VS cells connect amongst each other through electrical synapses. Further dye-coupling experiments also have shown that these electrical synapses were gap junctions [52]. In [37], they identified that these gap junctions are located at the axons of VS cells. Anatomically, these gap junctions are the only connection between the VS cells. Mechanistically, the VS network implements an electrotonic segregation through these gap junctions: all VS cells show broadened receptive fields at their axons compared to those at their dendrites. These broadened receptive fields improve the robustness of visual motion encoding [37, 73] at the output of the VS network. The upstream input from local motion detectors brings much more substantial synaptic conductance load on the dendrites than the dendritic-axonal leak, so placing a gap junction at the dendrites has negligible effects on the VS output [73]. However, placing gap junctions between the axonal compartments performs a kind of linear interpolation along the lateral (azimuth) direction [37]. This removes corruption of the ego-rotation signal arising from the inhomogeneous contrast distribution of the natural visual scene (and not from the

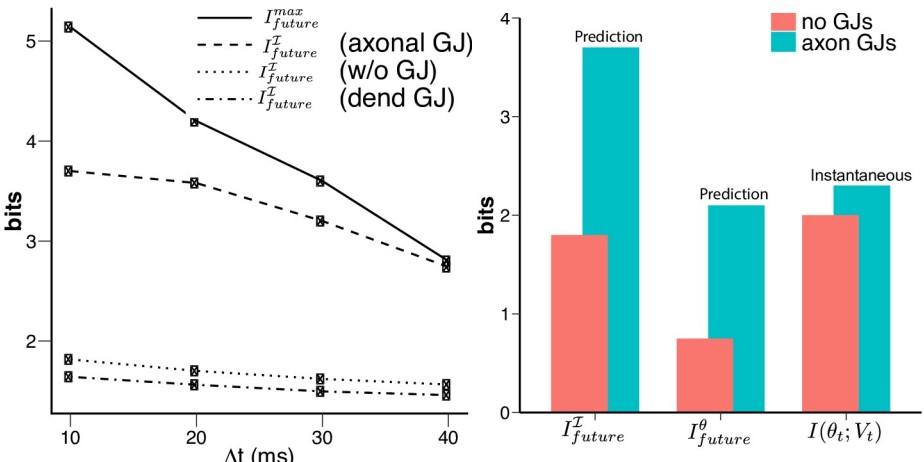

**Fig 2. The capacity of the VS network to encode predictive information varies with the anatomical locations of the gap junction between VS cells.** A) The predictive information about the future input current, $I_{\text{future}}$ encoded in four different schemes: 1) the past dendritic input current (solid line, this is the limit $I_{\text{future}}^{\max}$. It is also the upper bound of $I_{\text{future}}$), 2) the past axonal voltage when the gap junctions are present between VS axons (dashed line), 3) when the gap junctions are present between VS dendrites (dotted line) and 4) in the absent of gap junctions (dash-dotted line). All gap junctions = 1000 nS for both settings when they are present. Only their locations differ, i.e. axon vs. dendrite. Note that when the gap junctions are present between VS cell axons, the output voltages preserve almost the entire amount of the predictive information available at the inputs (red). (See also Materials and methods.) Such encoding is not because of linear correlation. As shown in, there is negligible linear correlation between the past and future egorotation at $\Delta t = 30 ms$ or $\Delta t = 40 ms$. B) The presence of axonal GJs improves the encoding of predictive information more than instantaneous input. We compare how the VS network encodes the predictive information in two scenarios (i.e. $I_{\text{future}}$ and $I_{\text{future}}(\theta, \Delta t)$ with $\Delta t = 10$ ms) and the instantaneous egomotion $\theta$ with axonal GJs (cyan bars) and without GJs (red bars). The encoding of instantaneous constant egomotion $I(\theta_t; V_t)$ (without prediction forward in time) is compiled from the previous work [46]. $I(\theta_t; V_t)$ was defined as the mutual information between a constant rotation $\theta$ and the transient axonal voltages of the VS network (integrated for $\Delta t = 10$ ms).

global ego-rotation signal, this is analogous to how dendritic integration at single VS cell removes corruptions along elevation, shown in [74]). Gap junctions also support a robust [37] and efficient [46] subpopulation readout scheme: the output from the VS network arises from subpopulations of adjacent cell triplets, which target different downstream areas [39, 75]. In particular, the VS network connects to the downstream descending motor neurons or neck motor neurons only through the VS 5-6-7 triplet of cells [50, 75], which have dendritic receptive fields located at the center of the fly's field of view. Therefore, these axonal gap junctions are dominant for network performance. Here, we investigate i) if axonal gap junctions improve prediction; ii) how such improvement is different from efficient coding of instantaneous input previously shown in [46].

Causality dictates that the past axonal voltages can only obtain information about the future current from the very own past current, therefore $I_{\text{future}}^{\max}$ (shown as the solid line in Fig 2A) is an upper bound on $I_{\text{future}}$. Here, we explore what network wiring features support the maximal transmission of the correlation structure in the input current onto the output axonal voltage of the VS network. As shown in Fig 2A, axonal gap junctions are necessary for the system to encode the maximal amount of predictive information about the input current. Namely, the $I_{\text{future}}^{\mathcal{I}}$ (shown as dashed line) only approaches $I_{\text{future}}^{\max}$ (shown as solid line) when gap junctions are present between neighboring VS axons. The other two configurations of gap junctions, i.e. no gap junctions or gap junctions at the dendrites (shown as dotted and dashdotted lines respectively), cannot encode as much predictive information.

We also observe that the presence of axonal gap junctions improves the encoding of predictive information much more than it improves the encoding of the instantaneous ego-motion [46] (Fig 2B). Comparing $I_{\text{future}}(\Delta t)$ with $I(\theta_t; V_t)$, we observe that the presence of axonal gap junctions improves $I(\theta_{t+\Delta t}; V_t)$ ($I^{\theta}_{future}$)and $I(\mathcal{I}_{t+\Delta t}; V_t)$ ($I^{\mathcal{I}}_{future}$) about 100% as opposed to 10% for $I(\theta_t; V_t)$; see the differences between the cyan and pink bars in Fig 2B. Because having axonal gap junctions can only reformat information received from the dendrites, this suggests that placing gap junctions at this position may be a strategy the VS network uses to specifically improve the predictive capacity of its output encoding. Next, we investigate whether such improvement approaches the physical limits set by the input current or ego-rotation information, computed using the information bottleneck framework.

## The VS network is near-optimal in predicting its own future input

All of the information encoded by the VS network comes from its sole input current, $\mathcal{I}_{\text{past}}$. To quantify the efficiency of encoding, we need to qualify not only the prediction (i.e. the $I_{\text{future}}$), but also how much the axonal voltages encode from past input $I_{\text{past}}$. Comparing $I_{\text{past}}$ and $I_{\text{future}}$, where the past is at time $t$ and the future at $t + \Delta t$, we can ask formally whether the VS network encodes as much as predictive information as possible, using the information bottleneck framework [76]. Given the amount of information the axonal voltage encodes about the past sensory input, what is the maximal amount of information it can retain about the future input? Such an optimum $I^{*}_{\text{future}}(\mathcal{I}, \Delta t)$ traces out a bound (the dark blue line) in Fig 3 as a

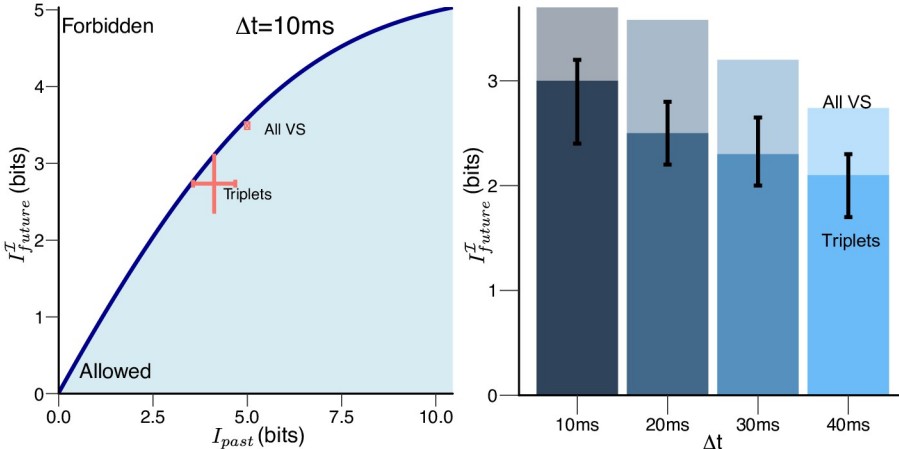

**Fig 3. Near-optimal prediction of the input to the VS network.** (A) The encoding of predictive information about the future current input to the VS network is near-optimal 10 ms after evasive maneuvers starts ($\Delta t = 10$ ms). Such performance is present for using both the entire VS network and the triplets. The dark blue curve traces out optimum encoding of future input to the VS network given varying amounts of information retained about the past input (also see Materials and methods). This curve also divides the plane into allowed (blue shaded region) and forbidden regions. No encoding can exist in the forbidden region because it cannot have more information about its future inputs than the input correlation structure allows, given causality and the data processing inequality. In addition, the maximal amount of information (shown as the highest point of the information curve) that is available as predictive information is limited by the correlation structure of the input (current), itself. We then plot the amount of information the axonal voltages of VS network (we show with axonal gap junctions in pink and without gap junctions in black) encode about the future input (the input current at time $t + \Delta t$) versus the information they retain about the past input (the input current at time $t$) (with all 120 triplets (crosses) and the whole network (circle)). The information efficiency, compared to the bound, contained in a particular encoding scheme corresponds to a single point in this ($I_{\text{past}}$, $I_{\text{future}}$) plane, which shows how much information it encodes about the past input vs. how much it encodes about the future. A particular VS encoding could occupy any point within the blue shaded region, but those that get close to the bound $I^{*}_{\text{future}}(\mathcal{I}, \Delta t)$ for a particular $I_{\text{past}}$ are the maximally informative predictors of the future input. (B) $I_{\text{future}}$ for all VS (light bars) vs. triplets (dark bars, with error bars) throughout the time span of evasive maneuvers ($\Delta t = 10$ ms, 20 ms, 30 ms, 40 ms).

function of $I_{\text{past}}$. It is the maximal possible predictive information at each level of compression, $I_{\text{past}}$. For encodings with the same $I_{\text{past}}$, those approaching the bound are optimal.

The known circuitry of the VS network allows us to probe two coupled questions: 1) What is the predictive efficiency (based on the encoding of the past) and 2) what is the predictive capacity (encoding of the past input only to predict the future input) of the VS network, given different readout circuitry (the entire VS network vs. the output VS 5-6-7 triplet)?

The predictive capacity of the VS network for its own future inputs is near-optimal. As shown in Fig 3A, the axonal voltages of the VS network encode $I_{\text{future}}$ = 3.49 ± 0.1 bits for future inputs at $\Delta t$ = 10 ms (the beginning of the banked turn). Considering that optimum is $I_{\text{future}}^*(\mathcal{I}, \Delta t) = 3.59$ bits, using axonal voltages of the entire VS network capture $I_{\text{future}}/I_{\text{future}}^*(\mathcal{I}, \Delta t) = 97.2\%$ of the optimal predictive information.

Similarly, using only the axonal voltages from the triplets, prediction of the entire VS network's future input is close to optimal as well (the cross in red in Fig 3A). All encodings based on outputs of triplets reach $I_{\text{future}}$ = 2.89 ± 0.36 bits while their respective physical limits are 3.07 ± 0.24 bits in Fig 3A. This suggests that all triplets achieve 89.8 ± 1.5% efficiency in encoding predictive information about the inputs $I_{\text{future}}/I_{\text{future}}^*$. VS triplets are not as efficient as the entire VS network in encoding the future dendritic input.

In general, triplets also retain less absolute predictive information about future input than the entire VS network throughout the time span of evasive maneuvers (Fig 3B). But the key point is not how well the triplet predicts the VS input, but how well it might help guide behavior by predicting the fly's future ego-rotation. We next explore this directly by asking how much information the triplet output has about the future ego-rotation.

## The triplet architecture selectively encodes predictive information about future ego-rotation

Here, we show that the triplet readout architecture retains nearly all of the available predictive information about the future ego-rotation $I_{\text{future}}(\theta, \Delta t)$ (light bars in Fig 4A) available to the VS network from the upstream input (darker bars in Fig 4A). Because downstream pathways of the VS network only read out from triplets, the VS network appears to use a two-step strategy to optimize this readout: it first efficiently represents correlations within its own past and future input, i.e $I_{\text{future}}(\mathcal{I}, \Delta t)$, at its output; then it selects components within that output that are relevant for predicting future ego-rotation, i.e. $I_{\text{future}}(\theta, \Delta t)$. This is possible because correlations coming from events in the visual world, such as the movement of large objects or the full-field background movement, have a different temporal structure (e.g. longer correlation times) than those internal to the brain.

We also observe that all triplets are near-optimal in encoding the predictive information about future ego-rotation (Fig 4B) (such optimality is also present for prediction of the distant future, i.e. $\Delta t > 10$ ms, results not shown). Considering that the VS 5-6-7 triplet encodes nearly as much information about the future ego-rotation as the VS network (Fig 4A), the main benefit of using triplets is compression: the VS triplet readout discards information predictive of the less behaviorally-relevant intrinsic dynamics of the inputs themselves. This compression is close to the niche region where the predictive information just begins to saturate, indicating that the triplet output has a better trade-off than the whole VS network in terms of how much $I_{\text{future}}(\theta, \Delta t)$ it can encode given its cost $I_{\text{past}}$.

Although all triplets encode similar amounts of information about the future ego-rotation (the standard deviation of the $I_{\text{future}}(\theta, \Delta t)$ amongst all 120 triplets is just 0.1 bits), the particular triplet connecting to the neck motor center, the VS 5-6-7, is one of the better choices in terms of how much information about the future ego-rotation it packs into its prediction of

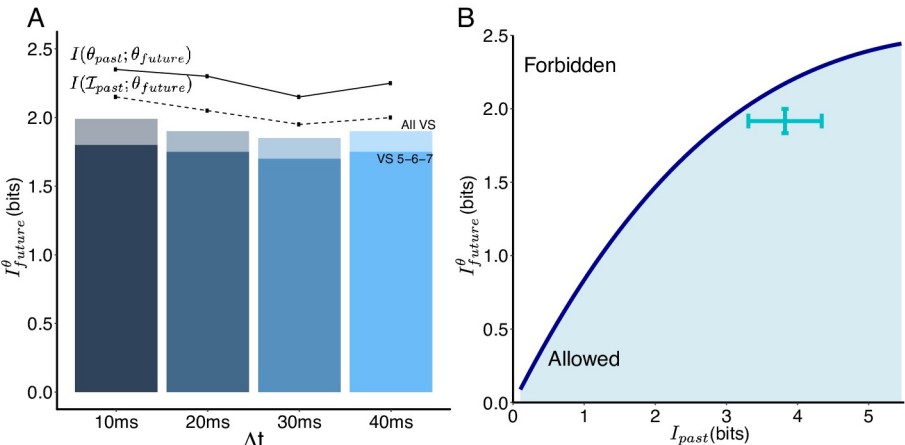

**Fig 4. Encodings based on the axonal voltages of triplets are near-optimal in predicting the future ego-rotation.**
(A) Histogram showing that the triplets (we use the output triplet VS 5-6-7 triplet here) encode nearly as much information *about the future ego-rotation* (shown in dark bars) vs. the entire VS network (shown in light bars), throughout evasive maneuvers. The solid line shows the mutual information within ego-rotation themselves: between the prior heading upon the detection of visual threat $\theta_{\text{past}}$ and different future ego-rotation throughout evasive maneuvers. The dashed line shows the mutual information between past input to the VS network and different future ego-rotation, all using the same past input $\mathcal{I}_{\text{past}}$ corresponding to the previous $\theta_{\text{past}}$. This is also the limit of prediction in the information bottleneck framework. (B) The encoding of predictive information for the $\theta_{\text{future}}$ at 10 ms after the start of evasive maneuvers ($\Delta t = 10$ ms). The dark blue curve traces out the optimum encoding for the future ego-rotation ($I(V_{past}; \theta_{\text{future}})$) given varying amounts of information retained about the past input ($I(V_{past}; \mathcal{I}_{\text{past}})$). The cyan cross corresponds to how much information each of all possible 120 triplets encodes about the future ego-rotation vs. how much information they retain from the past input.

the future input, while the VS 1-2-3 triplet is the most efficient. However, if we factor in wiring constraints, linking the output from VS 5-6-7 to a downstream dendritic arbor in the descending neurons for the neck motor center requires a much shorter wiring length compared to the peripheral location of the VS 1-2-3 triplet (VS cells are numbered according to their locations along the anterior-posterior axis; VS 5-6-7 are central in the compound eyes). It is possible that the minimization of wiring length [77] is important in selecting the simultaneously most predictive and most resource-efficient encoding.

Here we show that the VS 5-6-7 triplet that projects to the downstream neck motor center retains nearly all of predictive information about future ego-rotation as is present in the entire VS network. This result also shows that the predictive information encoded by the VS network is compressible: the VS 5-6-7 triplet successfully reformats the predictive information from 20 dendrites/axons (10 VS cells from both compound eyes combined) into six axons (the axons of VS 5-6-7 from both compound eyes combined). In the next section, we investigate how ego-rotation representations vary based on either the entire VS network or the VS 5-6-7 triplet. We hope to understand a) what kind of computation is possible via the encoding of near optimal predictive information, and b) how the VS 5-6-7 triplet reformats this near-optimal prediction.

## Predictive information encoded by the VS network provides fine-scale discrimination of nearby stimuli

Because the entire VS network has 20 VS cells and the VS 5-6-7 triplet only has 6 cells, it is difficult to directly compare their encoding of the fly's ego-rotation with different dimensions. However, both encoding schemes encode similar amounts of predictive information for ego-rotation. Because solutions with the same y-value or similar y-values on the information curve

share the same dimensionality in their respective compressed representations [76], the difference between these representations shows how the whole VS network differs from the VS 5-6-7 triplet in predictive information encoding. Because solving for these representation is generally intractable, we instead use the variational information bottleneck (VIB) to approximate these optimal representations [78–80]. The VIB is a generative learning framework. It is closely related to variational autoencoders [81]. Given pairs of inputs and outputs, it generates a latent feature space whose dimensions are predictive features from the input to the output (S6 Fig). One can then project the input into this latent feature space to obtain the predictive representation of the output. Here, we obtain these predictive representations in two steps: we first train the VIB to generate a latent feature space that maps the input (the axonal voltages of the VS network) to the future input current. We next project the axonal voltages (that correspond to the same future ego-rotation) onto this latent space. We can label these points in the latent space according to their respective future ego-rotation, and repeat this procedure for multiple ego-rotations. We can visually and computationally examine how overlapping or distinct these maximally predictive clusters of ego-rotations are in the latent space of the VIB. To allow for a direct comparison, we keep the dimension of the latent feature space to be the same while changing the input, using either the axonal voltages of the entire VS network, or those of the VS 5-6-7 triplet. S7 Fig shows that using two dimensional latent space already enables these predictive representations to encode substantial predictive information.

The VIB-generated representations of future ego-rotation (Fig 5) show that the predictive information encoded by the VS network supports fine-scale discrimination of the input motion direction. For ego-rotations with different degrees of clockwise roll and up-tilt pitch (but the same direction), the encoded predictive information puts those close in their azimuths far apart, i.e. a pair of ego-rotations (56° and 67°, shown in S9(A) and S9(B) Fig) with 10 degrees difference are mapped to distinct, well-separated clusters in the latent space of the VIB, whereas another pair (37° and 56°) that are farther apart share some overlap (S9(C) Fig). The VS 5-6-7 triplet preserves this fine scale discrimination (and S9(B) and S9(D) Fig) while compressing the readout. The same fine-scale discrimination is also present for ego-rotations combining counter-clockwise roll and up tilt, i.e. corresponding to vectors within the 4th quadrant of the fly's coronal plane (Fig 5B and 5D). However, these predictive representations cannot discriminate ego-rotations with vastly different roll or pitch directions, i.e. belonging to different quadrants: there is substantial overlap if we overlay these predictive representations, e.g. the cluster corresponding to 270° (shown in magenta in Fig 5C) will entirely cover the cluster corresponding to 19° (also shown in magenta, but in Fig 5A). The same overlap is also present in Fig 5B and 5D as well.

Even with a representation that retains all of ego-rotation-relevant information in the input to the VS network, one cannot use information available at the VS network input to discriminate ego-rotations of wide difference. We construct such a representation based on the instantaneous input current of the present ego-rotation. These input currents contain 2.44 bits relevant to the fly's instantaneous ego-rotation i.e. without prediction forward in time. This information is higher than that available via predictive information from the past input current (2.1 bits, shown as the red bar in S7 Fig). The first two principal components (PC) of the input current retain nearly all available ego-rotation relevant information, so we ask how ego-rotations disentangle in this representation of the first two instantaneous PCs. By projecting all VS input currents into these first 2 PCs, we find that there still exists substantial overlap between ego-rotation (S8 Fig), e.g. the cluster of 19° in magenta almost covers the entire cluster of 247° in light green (S8(A) Fig). This shows that the input to the VS network can only support fine-scale discrimination, whether an instantaneous readout or predictive readout.

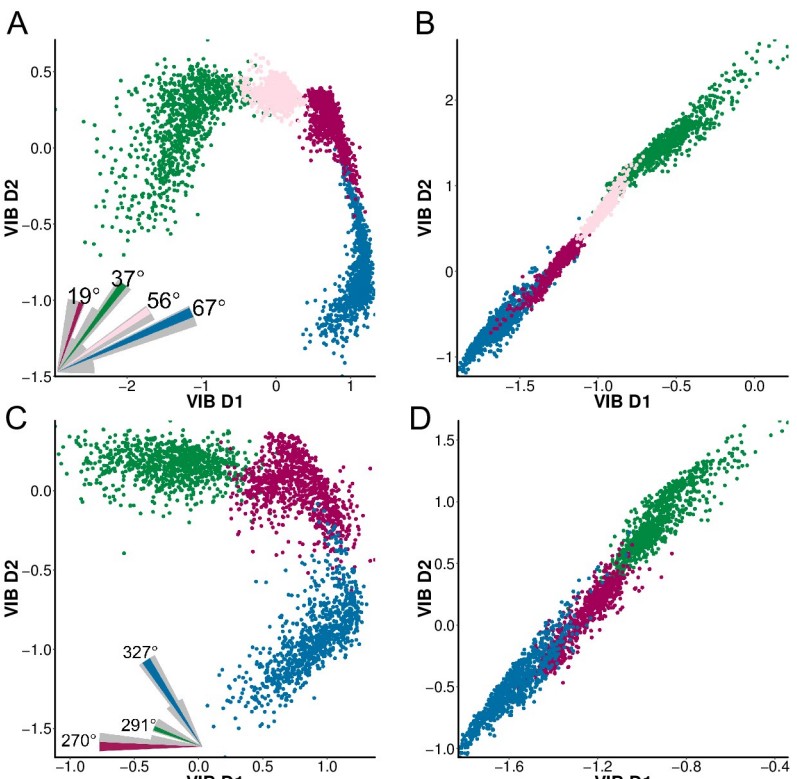

**Fig 5. The predictive information encoded by the VS network supports fine scale discrimination of future ego-rotation.** (A) The predictive representation of four future ego-rotations in the same quadrant of roll and pitch, e.g. an up-tilt and a clockwise roll. This representation maps the axonal voltage of the entire VS network to future ego-rotation through a latent feature space. The dimensions in this latent feature space (shown as VIB D1 and VIB D2) are VIB-learned predictive features based on the output of the VS network. All ego-rotation correspond to vectors within the 1st quadrant of the fly's coronal plane. The inset shows a polar histogram in grey and the four selected ego-rotations in color. (B) Similar to A but using the axonal voltages of the VS 5-6-7 triplet. (C) Similar to A, but ego-rotation are all counter-clockwise roll and up-tilt, corresponding to vectors in the 4th quadrant (between 270˚ and 360˚) of the fly's coronal plane. (D) Similar to C, but obtained using the axonal voltages of the VS 5-6-7 triplet as the VIB input. Note that although the overall correlation is high for the VIB solution using the axonal voltages of the VS 5-6-7 triplet, the VIB D1 and VIB D2 encode different information about $\theta_{future}$: VIB D1 encodes 1 bits about $\theta_{future}$ and VIB D2 encodes an additional 0.3 bits.

Ego-rotation representations are different if the system is optimized for prediction as opposed to optimized for encoding the present input. For example, the VIB-based predictive representation not only separates nearby ego-rotations (e.g. 56˚ and 67˚) into distinct clusters, but also inserts a cluster of another ego-rotation angle (e.g, 19˚) between them. By contrast, the disentanglement implemented by a circuit that maximally encodes the instantaneous inputs projects all distinguishable ego-rotations into adjacent clusters according to their azimuthal angle, i.e. 56˚ and 67˚ are next to each other. Because predictive representations contain less absolute information about the ego-rotation than the instantaneous-optimal representation, this difference suggests that the predictive information might preferentially support fine-scale discrimination of nearby ego-rotations. Information is not necessarily lost, however. Downstream, at the descending motor neurons [82–85] or the neck motor center [86, 87], information from other pathways (e.g. the haltere and prosternal organs [40]) is integrated. Thus, the discrimination of larger ego-rotation angles may be supported, while allowing the VS system to serve a specialized role in fine discrimination.

## Discussion

Here, by focusing our analysis of the fly's neural code for a key survival strategy, the evasive flight maneuver, we have shown that the VS network in the early fly visual system encodes behaviorally relevant predictive information near-optimally. A subpopulation readout mechanism, based on triplets of VS cells, further compresses the representation of that predictive information. This compression trades off the prediction of its own input with the prediction of future ego-rotation: while it encodes less absolute information about the future input, it retains the predictive information about what a fly will experience during evasive maneuvers at higher fidelity, in the outputs. The encoding of predictive information has a concrete behavioral goal: it enables fine-tuning of ego-rotation during evasive maneuvers.

Combining these observations, the fly brain satisfies an overarching computational goal of effectively guiding evasive flight trajectories through visual prediction at both levels of input filtering (via axonal gap junctions) and output reformatting (via subpopulation readout based on triplets). By next identifying that the predictive representations of future ego-rotation are best at enabling fine-scale discrimination of nearby ego-rotations, we have shown how structure maps to function in this motion sensing system. In addition, we have shown that behaviorally relevant features of ego-rotation are faithfully encoded via circuit mechanisms at the sensory end of the arc from sensation to action. This suggests that behavioral goals sculpt neural encoding even at the earliest stages of sensory processing.

Previous work had proposed that the halteres may implement a proportion-integration (PI) controller framework to generate motor commands [57]. In this framework, we hypothesize that feedforward visual control may have a functional role similar to the integration term: both are meant to reduce noise for high frequency signals, i.e. high velocity rotation in the evasive maneuver. A key difference between visual prediction and the integration term is that visual prediction does not merely respond to past errors, it gives an estimate of the future: it supports fine-scale discrimination of future ego-rotations at high velocity. This noise reduction is better suited for the abrupt, brief nature of evasive maneuvers, whose past error (before the initiation of evasive flight) may not be useful. Recent experiments [13] identified that visual information can shift the dynamic range of the motor neurons in the halteres. Namely, the halteres themselves obtain proportional gain for ego-rotation up to 2500˚/$s$ from its own mechonsensory input. The visual prediction then induces a shift in the haltere dynamic range so that they can output appropriate motor commands if a future ego-rotation is beyond 2500˚/$s$ and up to 5300˚. Whether this linear PI controller is indeed implemented by the halteres is an interesting direction for future research.

The halteres may implement the above PI controller in a different ways, some of them may predict novel sensory pathways between the visual system and a visually gated motor neuron of wing steering muscles. The halteres were known as a multi-sensory integration circuit. They combine inputs from the eye, ocelli, antenna and themselves using multiple campaniform sensilla embedded on the stalk. One possibility is that the descending visual input recruits the Coriolis-sensitive sensilla. This activates the reflex loop consisting of dF2 campaniform sensilla and wing steering motor neurons and thus directly modify the firing phase of the tonic first basalar muscle w.B1 whose phase advance associates with increased wingbeat amplitude [88]. Alternatively, the visual prediction may use dF3, a campaniform sensilla that receives input independent from dF2, to recruit the phasic motor neuron, i.e. the second basalare motor neurons M.b2 and steering muscle w.B2. Little is known about these neurons other than that they are responsible for initiating these elevated wing kinematics (stroke amplitude and frequency [54, 55, 89]) which are necessary for evasive maneuvers. In the prediction paradigm, when the predictive visual information tunes the strength of haltere mechanosensory

feedback [13], it indirectly activates these motor neurons via the haltere's connections to (M. b2) through electric synapses (fast enough for evasive maneuvers). Because of the lack of driver line targeting individual campaniform sensilla of the halteres, these are open questions for future investigation.

Our work also suggests that the encoding of predictive information is a key functional role of the lobula plate tangential neurons in dipterans. This might not be obvious given that different species have very different behavioral repertoires and selective pressures that sculpt their tangential neuron computations [40]. However, all dipterans use their lobula plate tangential neurons to encode motion information [40, 84]. All of these neurons have long processing lags, with respect to their behavioral timescale (e.g. the reaction time of robber flies are even faster for prey capture, around 10-30 ms, and their sensory processing delay is around 18-28 ms) [5]. The common theme that they overcome processing lags in their global motion sensitive neurons suggests that they may all use prediction to satisfy various selective pressure from different survival-critical behaviors. The predictive information from the visual system only accounts for about half of the future discriminability. We believe that the halteres, which integrate input from other sources, use the combination of visual feedforward prediction with other prediction to improve its motor command [88].

How the fly brain reads out the predictive information from the VS system is an important open question. Although the information contained by the temporally averaged output of the VS population response can be read out linearly, this may not be how readout takes place in the real downstream pathways of the VS network. For example, there are two descending DNOVS neurons (DNOVS1 and DNOVS2) connecting to the output from the triplet VS 5-6-7 [69, 75, 90, 91]. While DNOVS1 is a graded neuron, the DNOVS2 is a spiking neuron (with firing rates up to 100Hz). DNOVS2 introduces a substantial nonlinearity such that the biological readout may not be a simple integrator. In addition, the output of the VS network may be further combined with outputs from other LPTCs before they reach a motor control center [39].

Gap junctions are prevalent throughout the brain in many species [92, 93]. In vertebrate visual systems, the retina also encodes predictive information near-optimally to potentially circumvent sensory processing delays [18, 94]. Initial evidence supports the notion that gap junctions are a key circuit element in improving signal transmission in retina: for example, gap junctions between directionally selective ganglion cells in the mouse retina result in lag-normalization [95], and the gap junctions present in cones and bipolar cells improve the signal-to-noise ratio in their respective outputs [96]. Gap junctions can also rapidly regulate chemical synapses and improve sensitivity to correlated signals [97]. When processing stimuli with correlations between the past and the future (e.g. predictable motion), these mechanisms can support prediction to compensate for delays. In the central nervous system, gap junctions are versatile enough to support flexible hierarchical information processing in cortical circuits, as hypothesized in [98]. The ubiquitous evolutionary pressure to perform efficient prediction may shape nervous systems through this common circuit motif.

The brain carries out flexible, robust, and efficient computations at every moment as an organism explores and interacts with the external world. These computations are only possible through versatile mechanisms that operate under realistic behavioral constraints. We have shown that optimizing the transmission of predictive information in sensing systems is a useful way to interrogate the neural code. Given the presence of predictive information in sensory systems that evolved independently [18], our work supports the idea that predictive information may very well be a fundamental design principle that underlies neural circuit evolution. While we have dug into the specific mechanisms and representations that support this kind of

efficient prediction for fast, natural and behaviorally critical motion processing in the fly visual system, the lessons learned may apply to a much larger class of neural sensing systems.

## Materials and methods

### Ego-rotation for evasive flight maneuvers

We obtain ego-rotation stimuli from a large dataset of evasive flight maneuvers in *Drosophila* published in [21]. This dataset contains 82 traces of evasive trajectories when the flies face looming targets from all possible angles in their left visual field. All traces contain motion information (e.g. direction, velocity, etc.) from the emergence of the threat to the end of the evasive maneuver. In this dataset, the evasive flight trajectories are aligned at the beginning of the maneuver. In [21], they showed that both the speed and the expansion sizes of looming threats do not change the respective escape time courses and dynamics, i.e. this 40 ms evasive maneuver is the best *Drosophila* can do. The duration of the evasive trajectories vary between 10-40 ms, with 65 out of 82 flights as long as 40 ms. We chose this dataset for two reasons: a) its sample rate (7500 fps) allows us to trace the activity of the VS network at the millisecond scale; b) it contains threats approaching the fly from angles spanning a full 180°, providing a well-sampled collection of the fly's behavioral repertoire.

### Simulation of the in silico VS network

Our simulation uses a biophysically realistic simplified model of the VS network based on a reconstruction introduced in [37] (we used the modelDB python package introduced in [45]). This reconstruction models each VS cell with hundreds of dendritic compartments based on image stacks obtained by two-photon microscopy. Meanwhile, it implements the chain-like circuitry of the VS network by using both a) resistances connecting neighboring cells as axonal gap junctions [51, 52]; b) the negative conductance between the VS1 and the VS10 to account for the reciprocal inhibition [72].

Compared to the detailed reconstruction, the simplified, biophysically realistic model introduced in [38] reduces all dendritic compartments into a single compartment while keeping other components intact. In the simplified model, an individual VS cell is represented by one dendritic compartment and one axonal compartment, respectively. All its parameters were determined by a genetic algorithm [38] so that this simplified model behaves roughly the same as the real VS network when given the same current injection [51, 52]. Both the dendritic and axonal compartments have their own conductances ($g_{dend}$ and $g_{axon}$, respectively) and a connection conductance between them (shown as the $g_{dend/axon}$).

This simplified model does not include a few detailed dendritic structures of the VS cells, including the rotational distribution of dendritic branches and the newly identified inhibition from LPi [66] after this model was proposed. We don't think inclusion of these details would significantly change our results, especially given the consistency between this simplified model and real VS cells when having the same current injection [37, 39].

We combine the evasive traces and natural scene images to generate the optic flow patterns to which the VS network responds. For each of the 65 evasive traces that lasted a full 40 ms, we simulated 10,000 randomly generated natural scenes to obtain samples of the input (current arriving at dendrites) and output (axonal voltages) for subsequent analysis. In every simulation, we first generate the pseudo-3D visual "cube" (S1(A) Fig), representing the environment to which our model fly visual system responds, by randomly selecting six images from the van Hateren dataset. We then rotate this natural scene cube according to the rotational motion during evasive maneuvers recorded in [21] (we sample the rotational motion at a $\Delta t = 1$ ms interval, and integrate the VS response at a smaller time step of 0.01 ms to guarantee numerical

accuracy). This yields the optic flow pattern which we project onto a unit sphere that represents the fly's retina, following the protocol introduced in [39, 45].

The final step before simulating the VS network's response is to simulate its upstream local motion detection at the fly's retina. We use 5000 local motion detectors (LMD) evenly distributed on the unit sphere. Each LMD contains two subunits that differ by 2° in elevation. Their responses are $R(t) = (f * V_1)(t)(g * V_2)(t) - (g * V_1)(t)(f * V_2)(t)$ where $V_{1,2}(t)$ are photoreceptor responses at two neighboring locations and $f$ is the low pass filter. $g$ is the high pass filter [99].

Each VS dendrite takes as input the output of the LMDs that fall into its respective field. The receptive field (RF) of these dendritic compartments are modeled as 2-D Gaussian $S(x, y) = \frac{1}{2\pi\sigma_x\sigma_y} \exp\left(-\left[\frac{(x-x_{ctr})^2}{2\sigma_x^2} + \frac{y^2}{2\sigma_y^2}\right]\right)$ with azimuth $\sigma_x = 15°$ and elevation $\sigma_y = 60°$, tiling along the anterior-posterior axis (e.g. the centers of these receptive fields are located at $x_{ctr} = 10°, 26°, \cdots, 154°$ for VS1-10 respectively, see detailed configuration in [39]). The input current of an individual VS dendrite is the weighted sum of the synaptic conductance load from both excitatory and inhibitory inputs multiplied by the potential, i.e., with the excitatory conductance load as $(g_E = \Sigma_{x=-180°}^{180°}\Sigma_{y=-90°}^{90°} S(x, y)[R_t(x, y)]$ (similar for $g_I$, the input current is then $g_E * E_E + g_I * E_I$ ($E_{E,I}$ are reverse potentials relative to excitatory/inhibitory resting potentials, respectively). In our simulation, it is kept between -2.5nA to 2.5nA to be realistic [50]).

The neighboring axonal compartments of different VS cells are connected by gap junctions (shown as $g_{GJ}$), whereas VS1 and VS10 are connected by inhibitory chemical synapses. In our simulation, we set all conductance magnitudes using the same method as in [38]. Based on experimental findings [50], we vary the magnitude of the GJ conductance between 0 and 1 $\mu$S. Because the maximum firing rate of the descending neuron connecting to the neck motor center is 100 Hz [38], we use the transient temporal average of the resulting axonal voltage $\bar{V}_{past}^{tr} = 1/T \int V_{past}(t)dt$ (in the main text, the $V_{past}$ are temporal average. For simplicity, we use $V_{past}$ instead) for the subsequent analysis For the voltages just before the start of evasive maneuvers, we use the average from $t = -10 \sim 0$ ms, i.e. 0 ms is the start of evasive maneuvers.

## Efficient encoding of predictive information

To predict the future input motion, the only input the VS network has is its dendritic input at past times up to the present, i.e. $\mathcal{I}_{past}$. Ideally, the VS network output represents the future motion in a specific form, $Z$, following the optimal encoding dictated by the solution to our information bottleneck problem. The bottleneck minimizes how much the representation retains about the past input $I(Z; \mathcal{I}_{past})$ and maximizes how much it encodes about the future input i.e. $I(Z; \mathcal{I}_{future})$ (or $I(Z; \theta_{future}$ in Result). Formally, such encoding $Z$ solves the following variational problem, prediction of its own input:

$$\mathcal{L}_{p(Z|\mathcal{I}_{past}),\beta} = I_{past} - \beta I_{future} \tag{5}$$

where $\beta$ is the trade-off parameter between compression of information about the past, and retention of information about the future sensory input (we switch to $I_{future}(\theta, \Delta t)$ when we look at the prediction of future ego-rotation, as shown in Section 4 of the Result). For each $I_{past}$, there exists an optimal $I_{future}^*(I_{past})$ which is the maximum $I_{future}$ possible for a specified $I_{past}$, determined by the statistics of the sensory input, i.e. $\mathcal{I}_{past}$, itself.

We use the following iterative (the Blahut-Arimoto algorithm [100]) algorithm (the MATLAB implementation is available at: https://www.mathworks.com/matlabcentral/fileexchange/65937-information-bottleneck-iterative-algorithm by Shabab Bazrafkan to find $Z$

that optimizes Eq 5: (we use $X = \mathcal{I}_{\text{past}}$ and $Y = \mathcal{I}_{\text{future}}$ here)

$$p_t(Z|X) = \frac{p_t(Z)}{Z(X,\beta)} \exp[-\beta \sum_Y p(Y|X)) \log \frac{p(Y|X)}{p_t(Y|z)}] \tag{6}$$

$$p_{t+1}(Z) = \sum_X p(X)p_t(z|X) \tag{7}$$

$$p_{t+1}(Y|Z) = \sum_X p(Y|X)p_t(X|Z) \tag{8}$$

## Mutual information estimation

We use the *K*-nearest neighbor approach described in [101] (its open source software MILCA is available at https://www.ucl.ac.uk/ion/milca-0) to obtain mutual information estimates of $I_{\text{future}}(\mathcal{I}, \Delta t)$, $I_{\text{future}}^{\text{max}}$, $I_{\text{future}}(\theta, \Delta t)$ and $I_{\text{past}}$. Here, the mutual information is approximated via its corresponding complete gamma function:

$$I(X;Y) = \psi(K) - \ <\psi(n_x + 1) + \psi(n_y + 1)> + \psi(N) \tag{9}$$

*K* is the parameter for evaluating the complete gamma function $\psi$ and *N* as the sample size, here $N = 650,000$.

To choose *K*, we run the Blahut-Arimoto algorithm with a large $\beta$ (i.e. $\beta = 100$) to estimate the upper bound of mutual information, based on the observation that the information bottleneck reaches its optimum at $I(X;Y)$. Each Blahut-Arimoto estimation can take one week or longer in a multi-core cluster, so we only obtained the upper bounds for $I_{\text{future}}$, $I_{\text{future}}(\theta, \Delta t)$ and $I_{\text{past}}$ for the entire VS network and the VS5-6-7 triplet. We then use these upper bounds to determine *K*. In general, we use $k = 10, \cdots, 15$ (or $K = 1000, \cdots, 1100$ for those bootstrapped $\theta$ distributions) and calculate the mean as the estimate in our analysis. We omitted the standard deviations when we plotted Figs 2 and 4 because of their small magnitudes ($< 0.2$).

## Variational approximation of optimal encoding of the predictive information (VIB)

We use the variational approximation introduced in [78]. We first rewrite Eq 5 as Eq 10.

$$\mathcal{L}'_{p(z|\mathcal{I}_{\text{past}}),\beta'} = I_{\text{future}} - \beta' I_{\text{past}} \tag{10}$$

The minimization of Eq 5 is equivalent to the maximization of Eq 10 (i.e. when $\beta' = \frac{1}{\beta}$, Eq 10 is the same as Eq 5).

$$\mathcal{L}'_{p(z|\mathcal{I}_{\text{past}}),\beta} - \beta' H_{\theta_{\text{future}}} \geq$$
$$\mathcal{L}_{VIB} = \int dy dz\, p(\mathcal{I}_{\text{future}}, Z) \log q(\mathcal{I}_{\text{future}}|Z) \tag{11}$$
$$-\beta' \int d\mathcal{I}_{\text{past}} dz\, p(\mathcal{I}_{\text{past}})p(Z|\mathcal{I}_{\text{past}}) \log \frac{p(Z|\mathcal{I}_{\text{past}})}{r(Z)}$$

Next, we minimize the variational lower bound Eq 11 of Eq 10. The advantage of using this variational approximation Eq 11 is that we can constrain the distribution of *Z* to a particular form (i.e. a 2-D Gaussian) while letting the distributions of *x* and *y* to be arbitrary. This provides us with a latent feature representation of the lower bound for the optimal encoding of predictive information.

In this work, we would like to understand the structure of the optimal encoding for future ego-rotation given the input (the dendritic current, the VS axonal voltages, or the triplet voltages). Therefore, we obtain the respective solutions of $\mathcal{L}_{VIB}$ with fixed $\beta' = 40$. This is the value that falls into the diminishing return part of the IB curves in both Figs 3 and 4.

## Supporting information

**S1 Fig.** A) Schematic depiction of the visual stimuli for the simulation, recompiled from [46]. Six natural images (five are shown here, with one excluded to reveal the fly's viewing perspective) were randomly selected from the van Hateren dataset [65]; each image was patched onto a different face of a cube. Assuming that the fly is located in the center of this cube, we obtain the visual experience of the fly's ego-rotational motion by rotating this cage around a particular motion direction shown by the dark blue arrow. We then project the moving natural scene cage to a unit sphere that represents the fly's retina, following the protocol introduced in [39, 45]. There are ∼5,500 local motion detectors (LMD) evenly distributed on this unit sphere. The responses of those LMDs whose locations are within a VS cell's dendritic receptive field ($\Sigma_{azimuth} = 15$ and $\Sigma_{elevation} = 60˚$, tiling along the fly's anterior-posterior axis, see details in supplementary Materials and methods) are then integrated as the input current to this particular VS cell. B) Mercator maps with both checkerboard and natural scene backgrounds, at 1˚ resolution in spherical coordinates. C) ego-motion information inferable in checkerboard and natural scene backgrounds. The stimulus is a constant rotation of 500˚/$s$ from [46]. Note that there is only half of the information about this motion stimulus using the background of natural scene textures compared to the checkerboard background.
(TIF)

**S2 Fig. Egorotation distributions for different time steps during the evasive maneuver.** Egorotation distributions for different time steps during the evasive maneuver. Here we focus on the egorotations to which the VS network is sensitive. Because the VS network is only responsive to combinations of roll and pitch motions, i.e. motions within the fly's coronal plane, we represent all stimuli with their corresponding vectors in this plane. A) The egorotation distribution at 10ms before the onset of evasive maneuvers. B) The future egorotation at 10ms after the initiation of evasive maneuvers. C) Similar to B, but for the egorotation at 20ms within the evasive maneuver. Here, most of the banked turns slow down and counter banked turns start.) D) Similar to B, but for the egorotation at 30ms within the evasive maneuver. This motion corresponds to the start of the counter-banked turn. E) Similar to B, but for egorotations a fly would experience at the end of the evasive maneuver. This motion corresponds to the slowing down of counter-banked turn and the completion of evasive maneuver. All of these egorotation distributions have comparable entropy ∼4 − 4.3 bits. F) The Jensen–Shannon divergence between the past egorotation distribution and the egorotations at $\Delta t = 10, 20, 30, 40ms$ of evasive maneuvers, respectively.
(TIF)

**S3 Fig. Linear correlation between the past egoroation $\theta_{past}$ ($\Delta t = −10ms$ before the start of evasive maneuvers) and the future egorotation $\theta_{future}$ ($\Delta t = 10ms, 20ms, 30ms, 40ms$) at different time lags in the evasive maneuver.** These egorotations are calculated as the axis of rotation, combining the rotational angles along both roll and pitch body axes. A) The correlation between the egorotation distribution at 10ms before the onset of evasive maneuvers and the egorotation distribution 10ms into evasive maneuvers. B) Similar to A), but for the egorotation distribution 20ms into evasive maneuvers. C) Similar to A), the egorotation distribution 30ms into evasive maneuvers. D) Similar to A), the egorotation distribution 40ms into evasive

maneuvers.
(TIF)

**S4 Fig. Scatterplots of 120 triplets in A) the $I^{\mathcal{I}}_{future} - I_{past}$ plane; B) the $I^{\theta}_{future} - I_{past}$.**
(TIF)

**S5 Fig. How much a triplet based encoding retains from the past input vs. how much that information is about the future stimulus (out of the information about their own future input), for all 120 possible triplets.** The particular VS 5,6,7 triplet (shown by the red circle and the arrow) that connects with the neck motor center, is one of the most efficient in terms of how much fraction its prediction of its own input is about the future stimulus, while its encoding cost $I_{past}$ is modest.
(TIF)

**S6 Fig. Network schematic for the variational approximation of the information bottle-neck solution (VIB).** By constructing a variational approximation, the encoder learned a latent representation $\vec{z}$ from the past VS voltages. For training the encoder, we first project the axonal voltages of 20 VS cells to 200 intermediate filters, followed by a batch normalization layer. We then learn a latent representation with $z = 2$ for easy visualization. Then a decoder of the same structure as the encoder generates samples from $\vec{z}$ and reads them out as the future input current to the VS network. Note the VS network does not have direct access to the stimulus, it uses the correlations between its past and future inputs induced by the stimulus as a proxy for the stimulus correlations, themselves. $\vec{z}$ follows a Gaussian distribution, with parameters as $\mu$ and $\Sigma$. During training for this VIB, the mean $\mu$ and covariance matrix $\Sigma$ of $\vec{z}$ map the axonal voltages of VS to the future input. When the VIB succeeds, we obtain the predictive representation of the future stimulus by projecting their respective axonal voltages into the latent feature space of $\vec{z}$.
(TIF)

**S7 Fig. Predictive information for the future stimulus 10ms after the evasive maneuver starts ($\Delta t = 10ms$).** The red bar shows that the PCA projection of the first 2PCs from the input current contains almost all of the stimulus information available at the input current itself. We use this PCA projection to understand whether it is possible to disentangle input stimuli from different quadrants using prediction in Fig 5. The green bar shows the limit on prediction information, based on the information bottleneck method. It corresponds to the point on information curve at the given compression in Fig 4B. The cyan bar corresponds to the predictive information about the future stimulus using outputs from all VS cells. The darker-colored region shows how much information the corresponding VIB captures about the future stimulus. The purple bar is similar to the cyan bar, for predictive encodings of the VS 5-6-7 triplet vs. their respective VIB solution.
(TIF)

**S8 Fig. The input to the VS network only supports local discrimination.** A) The representation of 8 randomly selected stimuli within the plane whose dimensions are the first two principal components of the input currents. Note that there are substantial overlaps between clusters: e.g. the light-green cluster is almost on top of the dark-red/dark-blue clusters. B) The subset of 4 stimuli from A. The only difference, as compared to A, is that all these stimuli have the same pitch/roll directions (clockwise roll and up tilt pitch, i.e. they are all within the 1st quadrant of the fly's coronal plane).
(TIF)

**S9 Fig. The predictive information encoded by the VS network preferentially discriminates nearby egorotations.** A) The predictive representation of stimuli at 37˚ and 56˚ obtained by mapping the respective axonal voltages of the entire VS network to the latent feature space generated by the VIB. B) Similar to A, but using the VS 5-6-7 triplet as input. C) The predictive representation of two stimuli that are much closer in stimulus space: 56˚ and 67˚, respectively. Note that there is no overlap between these two nearby stimuli whereas there is some overlap for stimuli that are farther apart (shown in A). D) Similar to C, but using the VS 5-6-7 triplet as input.
(TIF)

## Acknowledgments

We thank D. Allan Drummond for graphical design of the 3D fly illustrations in Fig 1.

## Author Contributions

**Conceptualization:** Siwei Wang, Idan Segev, Alexander Borst, Stephanie Palmer.

**Data curation:** Alexander Borst.

**Formal analysis:** Siwei Wang.

**Funding acquisition:** Idan Segev, Alexander Borst, Stephanie Palmer.

**Investigation:** Siwei Wang.

**Methodology:** Siwei Wang, Stephanie Palmer.

**Project administration:** Idan Segev.

**Resources:** Idan Segev, Stephanie Palmer.

**Software:** Siwei Wang.

**Supervision:** Idan Segev, Alexander Borst, Stephanie Palmer.

**Visualization:** Siwei Wang, Alexander Borst, Stephanie Palmer.

**Writing – original draft:** Siwei Wang, Idan Segev, Alexander Borst, Stephanie Palmer.

**Writing – review & editing:** Siwei Wang, Idan Segev, Alexander Borst, Stephanie Palmer.

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
