## [Decision Letter · Decision Letter 0]

9 Nov 2020

Dear Dr Palmer,

Thank you very much for submitting your manuscript "Maximally efficient prediction in the early fly visual system may support evasive flight maneuvers" for consideration at PLOS Computational Biology.

As with all papers reviewed by the journal, your manuscript was reviewed by members of the editorial board and by several independent reviewers. In light of the reviews (below this email), we would like to invite the resubmission of a significantly-revised version that takes into account the reviewers' comments.

We cannot make any decision about publication until we have seen the revised manuscript and your response to the reviewers' comments. Your revised manuscript is also likely to be sent to reviewers for further evaluation.

Sincerely,

Lyle Graham

Deputy Editor

PLOS Computational Biology

Reviewer's Responses to Questions

**Comments to the Authors:**

Reviewer #1: Wang and colleagues present an analysis of numerical simulations of a realistic network of direction-selective VS cells in flies. The network is presented with the visual stimuli encountered during evasive maneuvers recorded in flying Drosophila, using a cube-geometry of natural images as visual inputs. The goal of this study is to understand whether signals in this network of neurons are predictive of future signals, during the evasive maneuver, such that they could be used for control during such fast maneuvers (typically on the order of 40 ms). The analyses appeared rigorous and well-done, but I thought some conceptual points could use clarification.

Major points

1) Line 46: Active control needs more definition. I interpreted it to mean on-going control during the evasive maneuver, rather than the motor activity stemming from a one-time command, and I’m not sure that the subsequent reasoning in this paragraph all supports that interpretation, since a one-time command could create points 1 and 4. Point 3 is just that all flight required on-going control, but not necessarily visual control. (And control on what timescale? Per wing-beat?) Point 2 seems most relevant, if flies update their escape maneuver in response to continuing changes in the looming stimulus during their maneuver. (Use of “active control” again in line 94).

2) Related to the point above: This whole analysis asks how present visual information can be used to predict future visual information, but if it’s being used to do this, isn’t that a forward model for control, rather than active control? This hinges on the author’s definition of “active control”, but it seems like ‘active control’ should be distinct from the kind of control that employs a forward model of what’s going to happen. Does the framework used here – in which current signals are informative about future ones -- differ from a forward model?

3) The authors note that these predictions are only possible because of correlations (not necessarily low-order) in the trajectory during these evasive maneuvers. A vanilla explanation of these results would be that simple, second-order trajectory correlations account for a good fraction of trajectory variance, and the VS network predicts the future well because it’s a good representation of the present. Can one ask how much variance in future state is explained by current trajectory? I guess I’m asking about whether some of this analysis can also be done using simple representations of temporal co-variance, and how different the results are when more sophisticated, information theoretic methods are used.

4) Input currents to VS cells are critical to define well, given their importance in this study: they were nowhere defined that I could see, though might be in a referenced paper. For instance, I expect these currents to include excitatory and inhibitory inputs from the LMD model, but do they allow those conductances to shunt current? For instance, if both excitatory and inhibitory conductances are high, is the current their sum, or an appropriately weighted sum reflecting the VS membrane voltage? Does it make a difference?

5) Related to the point above: how much do these results depend on the timescale of filtering of the local motion detectors? To obtain temporal frequency tuning of ~1 Hz (as in typical VS recordings), the delay line in any motion detector must do some substantial filtering over time, say 150 ms, which would cause reasonably long autocorrelations in the velocity signals. How does this affect the ability of this network to encode predicted changes in the future signal on timescales of the 40 ms maneuver? I can’t quite see how a long time delay in the motion detector could work with these short, fast maneuvers.

Minor points

1) Line 87: Not sure how an active counter-rotation differs from a counter-rotation.

2) Line 146: Citation for TF-tuning of local motion detectors only references theory for and recordings from an LPTC, which has spatially integrated, opponent-subtracted signals from local motion detectors. Creamer et al. 2018 showed that individual local motion detectors in Drosophila also have this tuning (before opponent subtraction).

3) Line 154: Add citations for LPLC2 and LC2. I don’t think the authors mean LC2 – it should be LC4. This will be Klapoetke et al. and Ache et al.

4) Fig 1C: Mercator projection seems to only show 180 degrees of azimuth. Perhaps scales would be helpful if a full 360 is not being shown?

5) Line 301: Talking about contrast heterogeneity. Not necessarily required for this model, but might be important to mention recent work in Drosophila on contrast normalization in the LMDs preceding LPTCs (Drews et al. and Matulis et al.). These sorts of effects almost certainly also exist in bigger Diptera.

6) Is the result in Figure 2 just due to averaging over space? Figure S3 seems to argue strongly against that, and the authors might consider moving that to a main figure.

7) Figure 3: Is it worth showing all triplets, rather than the cross bar? Would give a better sense of how many of them lie close to or far from the limit. Same with Figure 4B.

8) Since this is all information theoretic, is there a proposed method for the read out of the future information from the current state? Or just that it’s there?

9) Figure 5: numbers representing the angles are pretty unreadable. Please enlarge.

10) Figure 5BD: I don’t understand why these two VIB dimensions are so highly correlated. Does this mean a single dimension could do the VIB encoding in the case of this triplet?

11) Line 589: capitalize Drosophila

12) Line 634: need equation for local motion detection.

13) Line 638: reference V_past on both sides of equation, T is undefined.

14) Line 667: variable k appears undefined. Some capitalizations required in this paragraph.

15) Fig S2: Only 2 labeled panels, but don’t match the figure caption.

Reviewer #2: In this interesting paper the authors demonstrate that the VS network in the fly might be optimized to encode relevant information about future behavior during evasive flight manuevers. This work is a skillful combination of a broad range of approaches - natural stimulus and behavior statistics, detailed biophysical modelling and information theory. The paper generates novel experimental hypotheses, and provides a link between theoretical principles of efficient coding and predictive information and natural behavior. Overall, I think this work could be of potentially broad interest and relevance.

I have, however some concerns which I think the authors should address before acceptance.

1 - In reality evasive manuevers are triggered by a specific object in the visual field (e.g. an obstacle or a predator). Naturalistics scenes generated by the authors do not have, however such visual obstacles matched to the behavior - they are just images from the van Hateren database. That is the statistical structure of the scene and the evasive behavior are independent of each other. How does this affect the interpretation of the results? Could it be that after matching visual scene content to the behavior the ratio of I(theta, V) / S(theta) (Fig. 1B) would be much closer to 1? I think this point is central to the argument of the paper and should be explicitly discussed (and/or supported with additional analysis).

2 - The authors highlight the importance of the gap junctions (GJ) as a key biophysical component necessary to encode the predictive information (e.g. Fig. 2). These gap-junctions are a subset of parameters of the VS model. How many other parameters does the model have? Are there other parameters which might dramatically affect the network performance other than GJs? In other words - what makes GJs a unique subset of parameters from the perspective of predictive information coding?

3 - The idea of encoding the information about the organisms own future behavior is interesting, but could be perhaps better discussed. From a more "control-theoretic" perspective, the aim would be to encode the stimulus, incorporate it into the model of the environment, and then generate action which maximizes probability of the obstacle avoidance, given the flies current belief (or prediction) about the environment. Can one think of encoding the predictive information as extracting bits relevant only for such "control-theoretic" planning? I think it could be better explained and positioned in the context of the current literature.

4 - If I understood correctly, at best, predictive information is only arroud 50% of the entropy of the body rotation (Fig. 1B). If it is so - then to avoid the obstacle the flies still needs a lot of information. Where does it come from? How can partial information be used to perform the manuever much earlier? It would be good to discuss these aspects explicitly.

5 - The authors should dedicate more space to explain the relationship of this work to their previous paper [57]. In particular - if 57 claims that almost entire information about the rotation theta_t is encoded in the VS network state - how substantial is the current advancement? After all efficient coding and coding of predictive information will start to diverge when the bottleneck is strong (i.e. the network can retain only a small proportion of bits from the input). Even if my understanding of [57] is incorrect, this should be much more explicitly discussed.

6 - The article could be much more clear, and would definitely benefit from some streamlining. This work is a synergy of multiple approaches and research traditions - which is its strength. It however combines technical wording and explanations which make it confusing to readers who are not experts in all these fields (and I am clearly a member of this club). My specific suggestions are:

Shorten the introduction - it is very long and it is hard to understand what are the main contributions of the paper. Many parts of it could be moved to the results (e.g. description of the VS network)

There are very many information quantities with a lot of confusing indices. It took me a lot to map them all out, and I'm still not certain if I did it right. A clear diagram in Fig 1, explaining the relationship between V and I and I(V_past; Ipast), etc would be a great help. Fig 1. E does not seem to be enough.

Improve Fig 1 A - I find it very confusing - e.g. what does the dashed vertical arrow correspond to? Is it a process which happens instantaneously? Is all of the vertical dimension time?

**Have all data underlying the figures and results presented in the manuscript been provided?**

Reviewer #1: **No: **As far as I can tell, the numerical data underlying graphs is not available in spreadsheet form as supporting information. Although the authors reference lots of software packages to account for their simulations and fits, this work would be most reproducible if the analysis code were provided, perhaps also with intermediate data (like the output of simulations).

Reviewer #2: None

PLOS authors have the option to publish the peer review history of their article (what does this mean?). If published, this will include your full peer review and any attached files.

Reviewer #1: No

Reviewer #2: No
---

## [Decision Letter · Decision Letter 1]

20 Feb 2021

Dear Dr Palmer,

Thank you very much for submitting your manuscript "Maximally efficient prediction in the early fly visual system may support evasive flight maneuvers" for consideration at PLOS Computational Biology. As with all papers reviewed by the journal, your manuscript was reviewed by members of the editorial board and by several independent reviewers. The reviewers appreciated the attention to an important topic and your revisions. I am saying minor revision based on Reviewer 1's comments, which are truly minor, so I won't send it back out for review again.

Sincerely,

Lyle J. Graham

Deputy Editor

PLOS Computational Biology

[LINK]

Reviewer's Responses to Questions

**Comments to the Authors:**

Reviewer #1: In this revision, the authors have addressed well all the points I brought up. I think the revisions to the introduction improved clarity and I found the notation clearer as well. I'm also glad they will provide their code, since I think that will benefit the community.

A few minor notes:

Line 62: point 1 is a run-on sentence.

Line 55: I think it would be good to add a few citations to back up this claim about previous work.

Figure S3 could use x and y axis labels and units. I’m guessing these are in degrees per second. Is this roll rotation or yaw or total? Which rotation types are quantified/plotted is something that could be clarified throughout.

Line 285-288: Text does not quite match author’s description of it in response to minor point 5, since it cites only Drews, not Matulis.

Line 561: I’ve only ever heard these referred to as ‘campaniform sensilla’, never ‘campaniforms’.

Line 570: typo “the”?

Line 630: Capitalize?

Line 736: “Blahut-Arimoto” should be capitalized here and later in paragraph.

Reviewer #2: I thank the Authors for their response and modifications of the manuscript. In particular, I appreciate streamlining of the Introduction and the simplified notation of information quantities. The text is now much easier to understand (at least from my perspective).

I would encourage the Authors to include in the text some variant of their response to my question about the independence of the visual scene and the shape of the evasive trajectory (first question in the previous review). After all, this study connects statistics of stimuli to behavioral control, and many readers may wonder whether there is a link between the image of the obstacle/threat, and the evasive manuever.

Other than that, I think that the manuscript has now improved and will be of interest to a broad audience in computational neuroscience. I recommend it for publication, and congratulate the Authors.

**Have all data underlying the figures and results presented in the manuscript been provided?**

Reviewer #1: Yes

Reviewer #2: Yes

PLOS authors have the option to publish the peer review history of their article (what does this mean?). If published, this will include your full peer review and any attached files.

Reviewer #1: No

Reviewer #2: No

Figure Files:

Data Requirements:

Reproducibility:

References:

---

## [Editor Report · Decision Letter 2]

13 Apr 2021

Dear Dr Palmer,

We are pleased to inform you that your manuscript 'Maximally efficient prediction in the early fly visual system may support evasive flight maneuvers' has been provisionally accepted for publication in PLOS Computational Biology.

Best regards,

Lyle J. Graham

Deputy Editor

PLOS Computational Biology

---

## [Editor Report · Acceptance letter]

30 Apr 2021

PCOMPBIOL-D-20-01719R2 

Maximally efficient prediction in the early fly visual system may support evasive flight maneuvers

Dear Dr Palmer,

I am pleased to inform you that your manuscript has been formally accepted for publication in PLOS Computational Biology. Your manuscript is now with our production department and you will be notified of the publication date in due course.

With kind regards,

Andrea Szabo
